# Mis-Genotyping of Some Hepatitis D Virus Genotype 2 and 5 Sequences Using HDVdb

**DOI:** 10.3390/v12101066

**Published:** 2020-09-23

**Authors:** Caroline Charre, Frédéric le Gal, Paul Dény, Caroline Scholtès

**Affiliations:** 1University Claude Bernard Lyon 1 (UCBL1), University of Lyon, 69008 Lyon, France; caroline-m.charre@inserm.fr (C.C.); paul.deny@inserm.fr (P.D.); 2Laboratoire de Virologie, Hôpital de la Croix Rousse, Hospices Civils de Lyon, 69004 Lyon, France; 3Cancer Research Center of Lyon (CRCL), INSERM U1052, 69008 Lyon, France; 4Laboratoire de Microbiologie Clinique, Hôpitaux Universitaires de Paris Seine Saint-Denis, Site Avicenne, Université Sorbonne Paris Cité, 93022 Bobigny, France; frederic.legal@aphp.fr; 5Laboratoire associé au Centre National de Référence des virus des hépatites B, C et Delta, 93022 Bobigny, France; 6Unité INSERM U955, équipe n° 18, Université Paris Est, 94000 Créteil, France

**Keywords:** hepatitis delta virus, database, genotyping, webserver, errors, mis-genotyping

## Abstract

Evidence that Hepatitis D virus (HDV) genotype is involved in HDV infection pathogenesis is increasing. Indeed, HDV genotypes have been shown to be linked to different outcomes in terms of liver fibrosis and treatment response. Herein, we show that the promising HDVdb genotyping tool available online can lead to wrong genotyping results. The current HDVdb algorithm should be carefully considered as a “beta-version” and warrants algorithm core corrections, as soon as possible, for an optimal and beneficial use.

We read with great interest the manuscript “HDVdb: a comprehensive Hepatitis D Virus Database” by Usman et al. published in Viruses [1]. The authors developed highly useful tools available online (http://hdvdb.bio.wzw.tum.de/hdvdb/) to easily analyze HDV sequences by performing genotyping and phylogenetic trees. HDVdb is based on classic BLAST algorithm [2] with a dataset of references available online. We have recently developed a whole genome HDV long read sequencing combined with a dedicated local pipeline based on BLAST v2.9.0 and HDV reference sequences published on the International Committee on Taxonomy of Viruses (ICTV) website, https://talk.ictvonline.org/ictv-reports/ictv_online_report/negative-sense-rna-viruses/w/deltavirus.

Two output consensus sequences (dLy2020_1_Guinea_HDV-5, dLy2020_2_IvoryCost_HDV-5) that had been clearly affiliated to HDV genotype 5 (HDV-5) using our local bioinformatic pipeline were surprisingly assigned to genotype 6 (HDV-6) using HDVdb genotyping tool (http://hdvdb.bio.wzw.tum.de/hdvdb/bio/blast). In the HDVdb detailed output reports, maximum alignment scores were obtained with JA417556 and AX741159, HDV-6 reference sequences according to HDVdb. Other HDV-6 sequences namely JA417546, JA417551, AX741149 and AX741154 also showed high identity scores. JA417556, JA417546, and JA417551 from patent EP2343374 are identical to AX741159, AX741149 and AX741154 from patent WO03027291, and correspond to AJ584848|dFr910 (ICTV HDV-5 reference sequence), dFr47 and dFr73 strains, respectively. These sequences are the three original HDV genotype 5 reference sequences characterized [3]. As a consequence, when using the HDVdb genotyping tool, all HDV-5 sequences that obtain the highest BLAST-derived alignment score to one of these aforementioned sequences are affiliated with HDV-6, which is obviously wrong (Figure 1A). As suggested by the ICTV committee, these mis-labelled sequences should be relabeled in the HDVdb reference dataset as HDV-5 prototypes for a proper online genotyping tool.

We investigated results for the other genotypes with complete sequences (21 HDV-1, 7 HDV-2, 9 HDV-3, 7 HDV-4, 17 additional HDV-5, 12 HDV-6, 9 HDV-7, 8 HDV-8) accurately characterized by the French National Reference Center for HDV (FNRC, Bobigny, France) [4]. All tested sequences (*n* = 95) are available upon request. Further discordances were noted for HDV genotype 2 (HDV-2). Indeed, one HDV-2 sequence, namely LT604953, corresponding to the first HDV-2 described in Yakutia in 2001 (Strain “Yakut-6”) [5], was genotyped HDV-6 by the HDVdb genotyping tool (Figure 1B) and was also found in the overall complete genome dataset and the specific “HDV_genome_6” dataset provided online by HDVdb (Figure 1C, http://hdvdb.bio.wzw.tum.de/hdvdb/Downloads/Dataset/hdvdb_dataset_page/Genome_6.fasta). Here, again, a mistake concerning LT604953 affiliation was evidenced, which may lead to a wrong genotyping result for the query sequence. Additionally, three other HDV-2 sequences—namely X60193 (HDV-2 reference according to ICTV, as this strain “HDVJS” corresponds to the first HDV-2 strain described [6]), LT594481|strain dFr5820, and MA997213—were genotyped as HDV-1 by the HDVdb genotyping tool. LT594481|dFr5820 is also included in the “HDV_genome_1” reference dataset and may lead to further mis-genotyping of closely related sequences. In contrast, despite a wrong genotyping result as HDV-1 instead of HDV-2 for the X60193 sequence by the analysis tool (Figure 1D), the same accession number X60193 was correctly attributed to the “HDV_Genome_2” in the HDVdb reference dataset. Of note, the mis-attribution of genotypes in reference datasets may be due to the JAVA programming language pipeline created by Usman et al., to automatically extract for each accession number, the strain name, the genotype, the country of origin, and the date [1]. Of note, all other tested sequences were correctly genotyped.

As a confirmation, a phylogenetic tree, including the 10 potentially mis-genotyped sequences by the HDVdb online tool and 74 other sequences [7], was constructed using MEGA7 [8] after a multiple sequence alignment using MUSCLE [9]. The evolutionary analysis was inferred using the Neighbor-Joining method [10]. In the bootstrap test, 1000 replicate trees were computed. The evolutionary distances were computed using the Kimura 2-parameter algorithm [11]. A graphical viewer and tree figure drawing tool, Figtree v.1.4.4 (developed by A. Rambaut), was used to represent the phylogenetic tree in a cladogram type for better visibility (Figure 2). As is highlighted by the generated tree, the 10 sequences mentioned previously are clearly affiliated with their initial genotype according to Dény et al. [7] and the FNRC for HDV [4]. Thus, the phylogenetic tree reconstruction confirmed the false output results of the HDVdb genotyping tool (last accessed on 12 August 2020).

Genotyping of HDV may be crucial for the future management of HDV-infected patients, as it has been shown to be associated with the severity of liver disease as well as the treatment response [12,13]. However controversial results were recently published. For Spaan et al., HDV-5 seems to be associated with a favorable disease outcomes and a better PEG-interferon response compared to HDV-1 [14]. In contrast, Roulot et al. found that HDV-5-infected patients are at higher risk of developing cirrhosis, and that a successful response to PEG-interferon would depend more on the African origin of the patients than on the genotype [15]. Further investigations are necessary to clarify those results and establish the role of HDV genotype in treatment responses to new HDV therapies such as Bulevirtide, Lonafarnib or Rep3129 [16]. Therefore, online tools and public HDV sequence databases such as HDVdb are highly relevant and interesting for the HDV community to progress in HDV knowledge and to perform sequence analyses in an easier way. However, such databases have to be constructed very carefully, and will need to be regularly controlled and updated to avoid mistakes that may have a significant impact in terms of medical virology and clinical follow up. Of note, our check is obviously minimal and other mistakes could be discovered by using the platform. In conclusion, the current HDVdb genotyping tool algorithm should be carefully considered as a “beta-version”, and should be used with caution together with a classical phylogeny approach in order to correctly assess the HDV genotype from clinically-relevant HDV sequences. Mis-genotyping could lead to a false prognostic assessment for the patient and to inappropriate follow-up. Revision of HDVdb is mandatory to obtain correct genotypes, notably because it is essential to explore the association between the HDV genotype and the severity of HDV liver damage or therapeutic response.

## Figures and Tables

**Figure 1 viruses-12-01066-f001:**
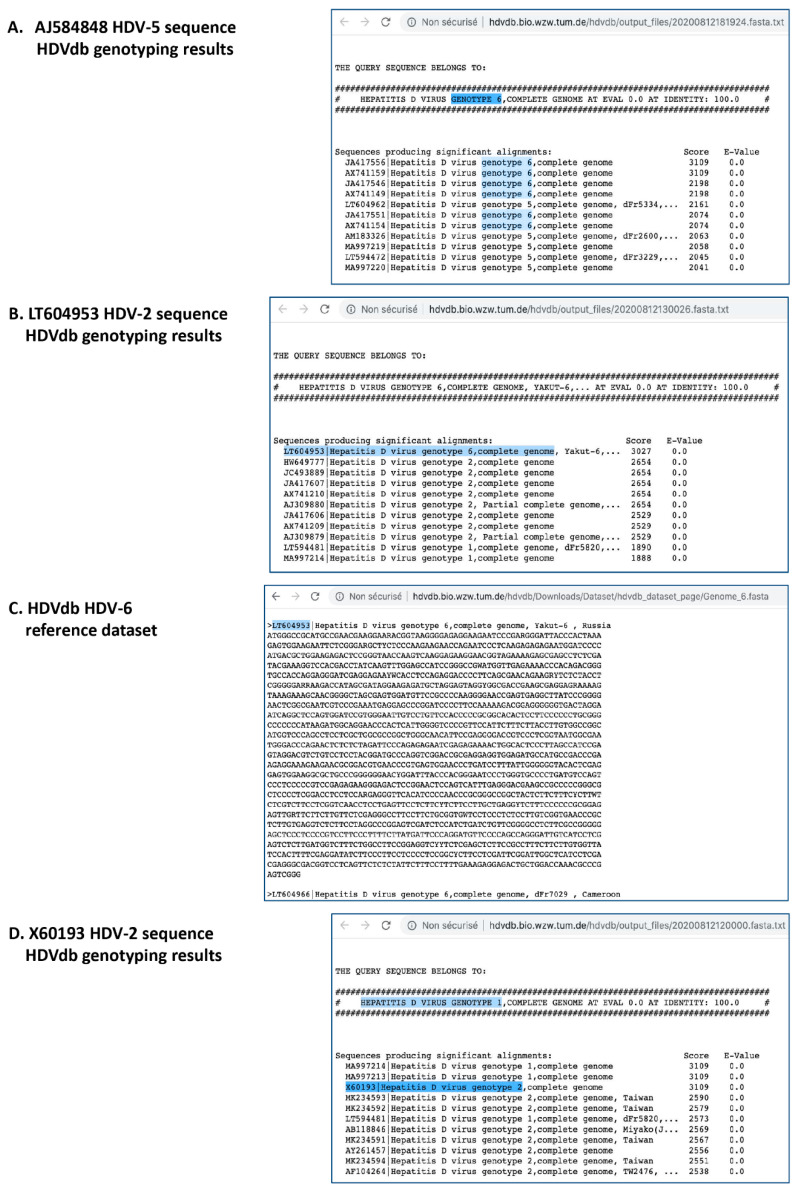
Screenshot of 4 different pages of HDVdb website. (**A**) HDV-5 ICTV reference sequence (AJ584848) genotyping output detailed results leading to HDV-6 query genotype. (**B**) HDV-2 sequence (LT604953) genotyping output detailed results leading to HDV-6 query genotype. (**C**) HDVdb HDV-6 complete genome reference dataset including an HDV-2 sequence (LT604953). (**D**) HDV-2 ICTV reference sequence (X60193) genotyping output detailed results leading to HDV-1 query genotype. Discrepancies are highlighted.

**Figure 2 viruses-12-01066-f002:**
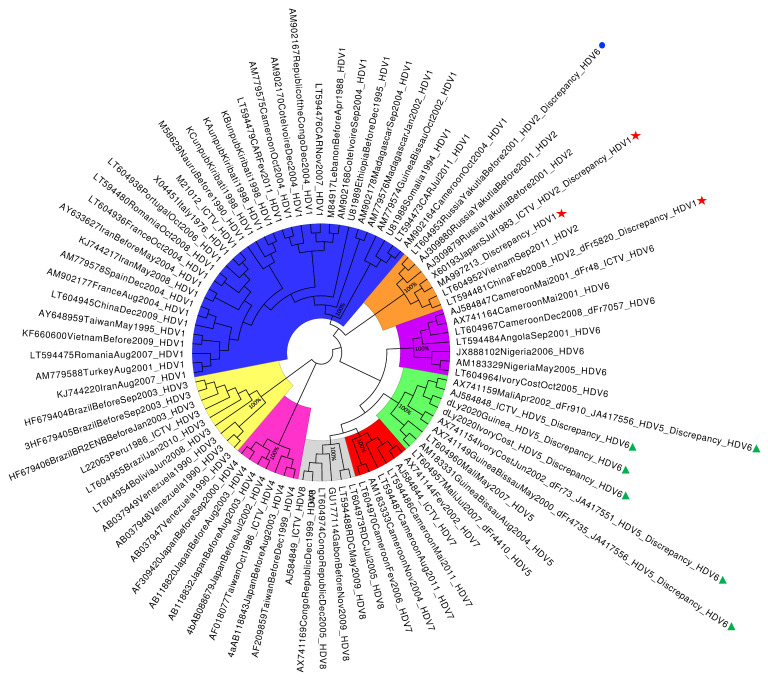
Phylogenetic tree computed with MEGA7 and represented upon a cladogram type using Figtree v1.4.4. The percentage of replicate trees in which the associated taxa clustered together in the bootstrap test (1000 replicates) are shown next to the main genotype branches. The analysis involved 84 complete genome sequences. All positions containing gaps and missing data were eliminated. There were a total of 1247 positions in the final dataset. HDV-1 is represented in blue, HDV-2 in orange, HDV-3 in yellow, HDV-4 in pink, HDV-5 in green, HDV-6 in purple, HDV-7 in red, HDV-8 in grey. Green triangles highlight HDV-5 sequences genotyped as HDV-6 by HDVdb genotyping tool. Red stars highlight HDV-2 sequences genotyped as HDV-1. Blue circle highlights HDV-2 sequence genotyped as HDV-6.

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
