# Peer review of "Mis-Genotyping of Some Hepatitis D Virus Genotype 2 and 5 Sequences Using HDVdb"

_viruses, 2020, doi:10.3390/v12101066_

Round 1
Reviewer 1 Report
This Letter comments on a recent article published in Viruses introducing a new Hepatitis D virus online set of tools that can be used to genotype HDV sequences and provides a downloadable set of HDV sequences. The authors have pointed out some important inaccuracies within the genotyping tool of the database, and related inaccuracies in the downloadable sequences. These are valid criticisms and it is important to let the wider viral hepatitis community know. The letter is well written, and I have no comments or suggestions for improvement.
Author Response
Ref.: Manuscript ID viruses-915658
"Letter to the Editor: Mis-genotyping of some Hepatitis D Virus genotype 2 and 5 sequences using HDVdb "
Dear Editor,
Please find appended a revised version of the manuscript entitled "Letter to the Editor: Mis-genotyping of some Hepatitis D Virus genotype 2 and 5 sequences using HDVdb" for publication in Viruses. We thank the Editor and the Reviewers for their comments that we have addressed in a point-by-point reply.
Yours sincerely,
Dr Caroline Charre, on behalf of the co-authors
Response to Reviewer 1 Comments
This Letter comments on a recent article published in Viruses introducing a new Hepatitis D virus online set of tools that can be used to genotype HDV sequences and provides a downloadable set of HDV sequences. The authors have pointed out some important inaccuracies within the genotyping tool of the database, and related inaccuracies in the downloadable sequences. These are valid criticisms and it is important to let the wider viral hepatitis community know. The letter is well written, and I have no comments or suggestions for improvement.
Authors’ response:
We thank the reviewer for this positive feedback.
Reviewer 2 Report
This is an important letter which demonstrates a deficiency with the HDVdb on-line database for HDV genotype analysis. 10 sequences were clearly mis-genotyped. The Science is valid and the letter is well-written. There are a couple of grammatical/typing errors that I have outlined below that would improve the fluency of this letter.
Lines 28-29: suggest removing “On our own” and beginning sentence with “We have recently…”
Line 45: remove “supposed to be” and write “are affiliated with HDV-6”
Line 57: add the word “a” to the end of the line – “lead to a wrong…”
Figure 1 title, line 73: I believe the sequence name should be LT604953, not LT604963
Line 96: Remove “a” and “point” – “may be crucial for…”
Line 97: change “to” to “with” – “associated with the severity…”
Line 99: change “to” to “with a” – associated with a favourable…”
Line 100: suggest changing “On the opposite” to “In contrast, Roulot…”
Line 101: suggest changing “satisfying” to “a successful response…”; reverse the order of the words “more” and “depend” – “would depend more..”
Line 109: suggest changing “non-exhaustive” to “minimal”
Line 113: Change “unappropriate to “inappropriate”
Author Response
Ref.: Manuscript ID viruses-915658
"Letter to the Editor: Mis-genotyping of some Hepatitis D Virus genotype 2 and 5 sequences using HDVdb "
Dear Editor,
Please find appended a revised version of the manuscript entitled "Letter to the Editor: Mis-genotyping of some Hepatitis D Virus genotype 2 and 5 sequences using HDVdb" for publication in Viruses. We thank the Editor and the Reviewers for their comments that we have addressed in a point-by-point reply.
Yours sincerely,
Dr Caroline Charre, on behalf of the co-authors
Response to Reviewer 2 Comments
This is an important letter which demonstrates a deficiency with the HDVdb on-line database for HDV genotype analysis. 10 sequences were clearly mis-genotyped. The Science is valid and the letter is well-written.
Authors’ response:
We thank the reviewer for this positive comment.
There are a couple of grammatical/typing errors that I have outlined below that would improve the fluency of this letter.
Lines 28-29: suggest removing “On our own” and beginning sentence with “We have recently…”
Line 45: remove “supposed to be” and write “are affiliated with HDV-6”
Line 57: add the word “a” to the end of the line – “lead to a wrong…”
Figure 1 title, line 73: I believe the sequence name should be LT604953, not LT604963
Line 96: Remove “a” and “point” – “may be crucial for…”
Line 97: change “to” to “with” – “associated with the severity…”
Line 99: change “to” to “with a” – associated with a favourable…”
Line 100: suggest changing “On the opposite” to “In contrast, Roulot…”
Line 101: suggest changing “satisfying” to “a successful response…”; reverse the order of the words “more” and “depend” – “would depend more..”
Line 109: suggest changing “non-exhaustive” to “minimal”
Line 113: Change “unappropriate to “inappropriate”
Authors’ response:
The manuscript has been revised accordingly. We thank the reviewer for outlining these grammatical/typing errors, and his suggestions to improve the manuscript.